# IGF2BP1 Significantly Enhances Translation Efficiency of Duck Hepatitis A Virus Type 1 without Affecting Viral Replication

**DOI:** 10.3390/biom9100594

**Published:** 2019-10-10

**Authors:** Junhao Chen, Ruihua Zhang, Jingjing Lan, Shaoli Lin, Pengfei Li, Jiming Gao, Yu Wang, Zhi-Jing Xie, Fu-Chang Li, Shi-Jin Jiang

**Affiliations:** 1College of Veterinary Medicine, Shandong Agricultural University, Taian 271000, Shandong, China; 2College of Public Health and Management, Weifang Medical University, Weifang 261042, Shandong, China; 3Shandong Provincial Key Laboratory of Animal Biotechnology and Disease Control and Prevention, Taian 271000, Shandong, China; 4Molecular Virology Laboratory, VA-MD College of Veterinary Medicine and Maryland Pathogen Research Institute, University of Maryland, College Park, MD 20742, USA; 5Department of Basic Medical Sciences, Taishan Medical College, Taian 271000, Shandong, China; 6College of Animal Science and Technology, Shandong Agricultural University, Taian 271000, Shandong, China

**Keywords:** duck hepatitis A virus type 1, insulin-like growth factor-2 mRNA-binding protein-1, 3′ untranslated region, translation efficiency, viral propagation

## Abstract

As a disease characterized by severe liver necrosis and hemorrhage, duck viral hepatitis (DVH) is mainly caused by duck hepatitis A virus (DHAV). The positive-strand RNA genome of DHAV type 1 (DHAV-1) contains an internal ribosome entry site (IRES) element within the 5′ untranslated region (UTR), structured sequence elements within the 3′ UTR, and a poly(A) tail at the 3′ terminus. In this study, we first examined that insulin-like growth factor-2 mRNA-binding protein-1 (IGF2BP1) specifically interacted with the DHAV-1 3′ UTR by RNA pull-down assay. The interaction between IGF2BP1 and DHAV-1 3′ UTR strongly enhanced IRES-mediated translation efficiency but failed to regulate DHAV-1 replication in a duck embryo epithelial (DEE) cell line. The viral propagation of DHAV-1 strongly enhanced IGF2BP1 expression level, and viral protein accumulation was identified as the key point to this increment. Collectively, our data demonstrated the positive role of IGF2BP1 in DHAV-1 viral proteins translation and provided data support for the replication mechanism of DHAV-1.

## 1. Introduction

Duck viral hepatitis (DVH), firstly described in Long Island in 1949 [1], is an acute and highly fatal disease of young ducklings characterized by severe liver necrosis, hemorrhage, and high mortality. This disease is caused by duck hepatitis virus (DHV) types 1, 2, and 3, and there were no antigenic relationships among them [2,3]. Duck hepatitis virus types 1 (DHV-1) was classified as a member of *Picornaviridae* and renamed as duck hepatitis A virus (DHAV) according to the Virus Taxonomy ninth Report of the International Committee on Taxonomy of Viruses (ICTV) [4]. Based on phylogenetic analyses and neutralization tests, DHAV was further classified into three serotypes: DHAV-1 (the classical serotype 1) [5], DHAV-2 (a serotype isolated in Taiwan) [6], and DHAV-3 (a serotype first identified in South Korea) [7]. Duck hepatitis A virus type 1 (DHAV-1) possesses a positive, single-stranded, non-enveloped RNA genome of ~7700 nucleotides (nt), including the 5′ untranslated region (UTR), one open reading frame (ORF), and 3′ UTR following with poly(A) tail [5].

Viral internal ribosome entry site (IRES) elements are highly structured RNA sequences that recruit the ribosome onto mRNA in a cap-independent manner. Internal ribosome entry site elements were initially reported in poliovirus (PV) and encephalomyocarditis virus (EMCV) genomes [8,9]. The IRES elements direct cap-independent internal translation initiation and canonical cap-dependent translation initiation [10]. Most eukaryotic mRNAs and many viral RNAs possess 5′ cap structure (m7GpppG), which could be recognized by translation initiation factor complex eIF4F (including eIF4E, eIF4A, and eIF4G) during translation progress [11]. For the non-capped viruses, the IRES elements possess diverse secondary structures with sizes ranging from 280 to 450 nt and can interact with cellular proteins for translation initiation [10]. The picornaviruses possess no 5′ cap structure at the 5′ terminus in contrast to cellular mRNA, and the initiation of viral protein synthesis was termed internal initiation that relies on the IRES element within 5′ UTR [12]. The IRES-mediated internal translation initiation is unique based on evidences that RNA structures of different IRES elements vary a lot, and their binding capacity for eukaryotic initiation factors and other RNA-binding proteins seems to be widely different [12,13,14]. For DHAV-1, the IRES element was categorized as a type IV IRES based on the secondary structure characteristics and biological properties and was found essential for the initiation of viral translation [15].

As the members of the Vg1 RBP/Vera, IMP-1,2,3, CRD-BP, KOC, ZBP-1 (VICKZ) family of RNA-binding proteins, insulin-like growth factor-2 messenger RNA (mRNA)-binding proteins (IGF2BPs) were also termed IGF-II mRNA-binding proteins (IMPs), zipcode mRNA-binding proteins (ZBPs), or coding-region instability-determinant binding proteins (CRDBPs) [16]. Insulin-like growth factor 2 mRNA-binding proteins (IGF2BPs), first identified in the human rhabdomyosarcoma cell line, interact with the human IGF-II leader-3 mRNA and regulate its translation [17]. Overall, IGF2BP1 is considered as a key regulator of mRNA metabolism with diverse roles in the control of mRNA localization, stabilization, and translation [17,18,19,20,21,22,23,24]. To hepatitis C virus (HCV), IGF2BP1 was found to strongly enhance IRES-mediated translation [23].

Up until now, little information is known about the replication mechanism of DHAV-1. Recently, we demonstrated that the 3′ UTR and the poly(A) tail could function as individual elements to enhance the DHAV-1 IRES-mediated translation [25]. In this research, we found that IGF2BP1 strongly enhanced DHAV-1 IRES-mediated translation efficiency through interaction with 3′ UTR. The exogenous expression of IGF2BP1 significantly increased the viral propagation, with no impact on the viral replication.

## 2. Materials and Methods 

### 2.1. RNA Pull-Down Assay

The RNA pull-down assay was performed using a Pierce magnetic RNA–protein pull-down kit (Thermo Fisher Scientific, Waltham, MA, USA) according to manufacturer instructions. The entire 3′ UTR with poly(A)_25_ tail, the reverse complementary sequence the of 3′ UTR with poly(A)_25_ tail, the 3′ UTR alone, or the poly(A)_25_ tail alone in vitro transcribed RNA samples were prepared using the T7 RiboMAX Express large-scale RNA production system (Promega, Madison, WI, USA), followed by biotin-labeling at the 3′ termini of the RNA samples using a Pierce RNA 3′-end desthiobiotinylation kit (Thermo Fisher Scientific) according to manufacturer instructions. The biotin-labeled RNA samples were then used in combination with magnetic beads. The duck embryo epithelial (DEE) cell line [26] infected with DHAV-1 was isolated at 48 h postinfection (hpi) and centrifuged at 3000×g for 5 min at 4 °C, and the collected cells were lysed using 100 µL of passive lysis buffer (Promega). Thereafter, the cell lysates were added to the magnetic beads–RNA–biotin mixture and the isolated proteins were subjected to SDS-PAGE analysis, followed by the silver-staining using SilverQuest™ Silver Staining Kit (Thermo Fisher Scientific) according to manufacturer instructions. The pulling-down product was subjected to mass spectrometry.

### 2.2. Plasmids

The monocistronic reporter plasmid pDHAV-3′UTR-A_25_ contained the following elements from 5′ to 3′ in a pBluescript II KS (+) vector: the cytomegalovirus (CMV) immediate early promoter, a T7 promoter, the entire 5′ UTR (nucleotides 1–626) of DHAV-1, and the Firefly luciferase (FLuc) gene, followed by DHAV-1 3′ UTR and poly(A)_25_ tail. Mutated monocistronic reporter plasmids absent of the 3′ UTR, the poly(A) tail, and both the 3′ UTR and the poly(A) tail were named pDHAV-Δ3′UTR-A_25_, pDHAV-3′UTR-ΔA_25_, and pDHAV-Δ3′UTR-ΔA_25_, respectively (Figure 1A). The T7 promoter and Ranilla luciferase (RLuc) gene were inserted into pBluescript II KS (+) vector to establish plasmid pRluc (Figure 1B). The RNA-launched infectious clone of DHAV-1 named pR-DHAV-1 (Figure 1C) was previously established based on the strain LY0801 (accession no. FJ436047) [27].

The recombinant plasmid pBlue-Ribo-DHAV-1 was established by inserting DHAV-1 5′ UTR, a hammerhead ribozyme sequence, polyprotein sequence of DHAV-1, hepatitis delta virus ribozyme sequence, and 3′ UTR plus poly(A)_25_ tail of DHAV-1 into pBluescript II KS (+) vector (Stratagene, La Jolla, CA, United States) (Figure 1D). The hammerhead ribozyme and hepatitis delta virus ribozyme could lead to self-cutting to enrich the viral copy numbers without expression of viral proteins under the condition of existing Mg^2+^ or other divalent ions and a cytidine nucleotide. The recombinant plasmid named pBluescript-polyprotein was constructed by inserting the polyprotein coding sequence of DHAV-1 (base position of 627–7376) into pBluescript II KS (+) vector for expression of the polyprotein of DHAV-1 (Figure 1E). The IGF2BP1 coding sequence was cloned into pLVX-Tight-Puro vector (Clontech, Palo Alto, California, USA) and was named pLVX-Tight-Puro-IGF2BP1. Moreover, the IGF2BP1 coding sequence was cloned into pcDNA™3.1/V5-His A vector to generate recombinant plasmid pcDNA-IGF2BP1.

### 2.3. In Vitro Transcription

The complete or mutated RNA-launched infectious clones of DHAV-1 were linearized with XhoI and then gel purified according to manufacturer instructions (Omega Bio-Tek, Norcross, GA, USA). The gel-extracted products were quantified using a spectrophotometer (Eppendorf, Cambridge, UK), followed by Sp6 polymerase-mediated in vitro transcription (Promega). The monocistronic reporter plasmids, bicistronic reporter plasmids, or pBlue-Ribo-DHAV-1 were digested with XhoI and then gel purified. The gel-extracted products were then used for in vitro transcription using the T7 RiboMAX Express large-scale RNA production system (TaKaRa, Dalian, China).

### 2.4. Cells and Transfection

DEE cells or HEK 293T cells (ATCC CRL-11 268) were cultured at 37 °C in 5% CO_2_ in Dulbecco’s modified Eagle medium (DMEM; Gibco, Carlsbad, CA, USA) supplemented with 10% fetal bovine serum (FBS, Gibco), 100 U/mL penicillin, and 100 µg/mL of streptomycin sulfate. For liposome-mediated transfection, cells were seeded into 24-well plates and grown to 60% confluence. For RNA transfection, 0.8 µg of the in vitro transcribed RNA was diluted in 50 µL opti-MEM (Thermo Fisher Scientific) and 2 µL (for 24-well plate) of RNA transfection reagent (TIANGEN, Beijing, China) was mixed in another 50 µL opti-MEM, following with 5 min of incubation at 25 °C. The two reagents were then mixed together and incubated at 25 °C for another 25 min; the mixture was then added to DEE cells. Then, 0.1 µg of pGMLR-TK Renilla Luciferase Reporter Plasmid (Promega) was co-transfected with reporter RNAs and the RLuc activity was detected to guarantee consistency of the RNA transfection efficiency among all groups.

### 2.5. In Vitro RNA Pull-Down

pcDNA-IGF2BP1 (Figure 1F) was transfected into HEK 293T cells to express IGF2BP1-His fusion proteins. Cell culture was collected at 48 h post transfection (hpt) and then purified by Ni^2+^ affinity chromatography His-Bind Resin (Novagen, Madison, WI, USA) according to the manufacturer’s instructions. In brief, cells expressing IGF2BP1-His fusion protein was then sonicated on ice, and cell lysates were then purified on a gravity-flow column packed with 3 mL Ni^2+^-NTA resin slurry (Novagen, Madison, WI, USA). The His-tagged proteins were eluted in 50 mM NaH_2_PO_4_, 300 mM NaCl, and 300 mM imidazole (pH 8.0). The purified IGF2BP1-His fusion proteins were then used for RNA pull-down assay as mentioned above.

### 2.6. Establishing a Stable DEE Cell Line Continuously Expressing IGF2BP1

The HEK 293T cells were seeded in the 6-well plate and cultured until the cells reached 60% confluence. Lentiviral particles were produced in the HEK293T cells with the Lenti-X Packaging System (Clontech) following the manufacturer’s instructions. Briefly, 37 µL of Lenti-X HT Packaging Mix, 6 µL (about 4.0 µg) of pLVX-Tight-Puro-IGF2BP1 plasmids, and 557 µL of Xfect Reaction Buffer were put together and then mixed with 7.5 µL of Xfect Polymer diluted in 592.5 µL of Xfect Reaction Buffer. The mixture was gently mixed and then incubated for 10 min at 25 °C, followed by addition to the 6-well plate. At 10 hpt, the mixture was replaced by DMEM supplemented with 10% FBS. At 72 hpt, the supernatant was collected to infect DEE cells. At 24 hpi, the DEE cells were cultured in DMEM supplemented with 5% fetal bovine serum, 500 μg/mL G418, 2 μg/mL puromycin, and 100 µg/mL of streptomycin sulfate. After approximately 17 days, the DEE cell line that stably expressed IGF2BP1 was established.

### 2.7. RNA Interference (RNAi)

Duck embryo epithelial cells (~5 × 10^5^ cells in DMEM without FBS and antibiotics) were transfected with 1.25 µL siRNA (CCA ACA ACC CGT TGT ACA A) (20 µM) and 3 µL riboFECTTM CP Reagent (RIBOBIO, Guangzhou, China) according to the manufacturer recommendations. Cell lysates were collected from 24 to 72 hpt and were then lysed in radio immunoprecipitation assay (RIPA) buffer (50mM Tris-HCl at pH 7.5, 150 mM NaCl, 1% Ipegal, 0.5% sodiumdesoxycholate, and protease inhibitors) for 15 min on ice following 10 min of centrifugation at 10,000×g at 4 °C. Protein concentration was then determined by bicinchonininc acid (BCA) assay (Thermo Fisher Scientific). The lysis was then used for subsequent gene expression analysis through western blot.

### 2.8. Luciferase Activities Measurement

At different time points after transfection, the medium was discarded and the cells were washed three times with phosphate-buffered saline (PBS). Following that, 100 µL of passive lysis buffer (Promega) was added into each well and the plate was put on a horizontal table for 5 min at room temperature. To measure the FLuc activities, cell lysates were collected and centrifuged at 13,000×g for 5 min at 4 °C and the pellet was discarded. Luciferase activities were measured in 50 µL aliquots of lysed cells using a dual luciferase assay system (Promega) and a Berthold luminometer (GloMax 20/20; Promega).

### 2.9. Western Blot and Antibodies

Cell lysates were subjected to sodium dodecyl polyacrylamide gel electrophoresis (SDS-PAGE) on a 12% polyacrylamide gel, and the separated proteins were electroblotted onto a polyvinylidene fluoride (PVDF) membrane (Thermo Fisher Scientific). Thereafter, the PVDF membrane was blocked with 5% non-fat milk in Tris-Buffered Saline Tween-20 (TBST) (500 mL NaCl, 0.05% Tween 20, 10 mM Tris-HCl, and pH 7.5) for 1 h. The membrane was incubated at 4 °C for 8 h with anti-DHAV-1 monoclonal antibody (mAb) 4F8 (1:500) [28], mouse anti-FLuc mAb (Abcam, Cambridge, MA, USA, 1:3000), mouse anti-RLuc mAb (Abcam, 1:3000), mouse anti-Renilla Luciferase mAb (Abcam, ab82968, 1:3000), mouse anti-His-tag mAb (CoWin Bioscience, Beijing, China, 1:3000), and mouse anti-IGF2BP1 mAb (Abcam, 1:3000). The membrane was washed four times with TBST and incubated with horseradish peroxidase (HRP)-conjugated goat anti-mouse antibody (Abcam, 1:3000) at 4 °C for 4 h. The PVDF membrane was then visualized with hydrogen peroxide and 3,3′-diaminobenzidine tetrahydrochloride (Sigma-Aldrich, St. Louis, MO, USA).

### 2.10. Immunofluorescence Assays

The DEE cells were rinsed with PBS, fixed with 0.2% polyoxymethylene for 10 min at room temperature, and then incubated with mouse anti-IGF2BP1 mAb (Abcam, 1:3000) for 1 h at 37 °C. After reaction with fluoresceine isothiocyanate (FITC)-conjugated goat anti-mouse antibody (Abcam, 1:3000), the samples were washed three times with PBS and inoculated with 500 μL 4′,6-diamidino-2-phenylindole (DAPI; Merck, Germany) for 10 min. The slips were then visualized by laser scanning confocal microscopy (Leica AF6000).

### 2.11. Quantitative Real Time-Polymerase Chain Reaction

The extracted RNA was quantified by quantitative real time-polymerase chain reaction (RT-qPCR), as described previously [29]. Using the primers RT-qPCR-F/R (F: 5′-AGA CAC ATG TTG CTG AAA AAC T-3′, R: 5′-TCC CCC TTA TAC TTA ATG CCA G-3′) and the DHAV-1-Probe Cy5-5′-ATG CCA TGA CAC TAT CTC ATA TGA GTC AGC-3′-BHQ-2, a TaqMan real-time RT-PCR assay for quantitative detection of DHAV-1 was conducted. Viral RNA copies were calculated using the formula X = 6.7 × 10 (40.812 − y)/3.285, where X represents a standard of viral copies and y represents a standard of values derived from one-step real-time PCR.

### 2.12. Statistical Analyses

All experiments were performed at least three times with at least three biological replicates. Statistical significance was evaluated by statistical analysis system (SAS) software (v.8.2; SAS Institute, Inc., Cary, NC, USA) to mean ± SD using the two-tailed Student′s test (**p* < 0.05, ***p* < 0.01, and ****p* < 0.001).

## 3. Results

### 3.1. IGF2BP1 Specifically Interacted with 3′ UTR of DHAV-1

To explore the functional proteins that specifically interacted with DHAV-1 3′ UTR and poly(A) tail during viral propagation, RNA pull-down assay was performed based on these two functional elements. The four labeled RNA probes used in the RNA pull-down assay include the entire 3′ UTR of DHAV-1 with poly(A)_25_ tail (probe 1), the reverse complementary sequence of the 3′ UTR of DHAV-1 with poly(A)_25_ tail (probe 2), the 3′ UTR of DHAV-1 alone (probe 3), and the poly(A)_25_ tail alone (probe 4). The isolated proteins of the four groups were analyzed by the silver-staining electrophoresis, followed by Matrix-assisted laser desorption/ionization time-of-flight mass spectrometry (MALDI-TOF-MS) analysis. The MALDI-TOF-MS results showed that protein bands marked with red arrows (Figure 2A) in probes 1 and 3 groups include a common component IGF2BP1, and IGF2BP1 processed the highest score in both groups, suggesting that the protein might be involved in the viral propagation (Figure 2B). Thus, IGF2BP1 was selected as a target for further analysis. In order to identify whether IGF2BP1 directly interacted with 3′ UTR, in vitro RNA pull-down was conducted with purified IGF2BP1-His fusion protein and four groups of probes. The isolated proteins were analyzed through western blot with anti-IGF2BP1 and anti-His, and the results showed that IGF2BP1 specifically interacted with probes 1 and 3, which further confirmed the MALDI-TOF-MS results (Figure 2C). These data indicated that IGF2BP1 directly interacted with 3′ UTR of DHAV-1.

### 3.2. IGF2BP1 Strongly Enhances DHAV-1 IRES-Mediated Translation through Binding to 3′ UTR

The DEE cell line stably expressing IGF2BP1, named ExIGF2BP1 DEE, was successfully established using the lentiviral system. Moreover, RNAi assay against IGF2BP1 was performed to downregulate IGF2BP1 expression levels in DEE cells (named SiIGF2BP1 DEE). Western blot result showed that IGF2BP1 expression levels significantly decreased in the SiIGF2BP1 DEE cells and increased in the ExIGF2BP1 DEE cells (Figure 3A).

In order to study the role of IGF2BP1 during DHAV-1 propagation, the reporter RNAs of pDHAV-3′UTR-A_25_ and pDHAV-Δ3′UTR-ΔA_25_ were respectively transfected into the ExIGF2BP1 and SiIGF2BP1 DEE cells and FLuc activity was measured at 4 hpt. In order to measure the RNA transfection efficiency among each group, equal copies of in vitro transcribed products of pRluc were also transfected into these DEE cells as the control. The results showed that the FLuc activity in the pDHAV-3′UTR-A_25_ group significantly (*p* < 0.001) increased in the ExIGF2BP1 DEE cells or significantly (*p* < 0.001) decreased in the SiIGF2BP1 DEE cells, while the FLuc activity in the pDHAV-Δ3′UTR-ΔA_25_ group showed no obvious difference (*p* > 0.05) in the different IGF2BP1 expression level DEE cells (Figure 3B). Meanwhile, the DHAV-1 RNA-launched infectious clone pR-DHAV-1 was transfected into the ExIGF2BP1 and SiIGF2BP1 DEE and the cell lysates were collected from 24 hpt to 60 hpt to conduct western blot assay with anti-DHAV-1 mAb 4F8. The results showed that upregulated IGF2BP1 expression in DEE cells significantly increased the viral protein expression level of DHAV-1, while knockdown of IGF2BP1 significantly reduced the viral translation efficiency at each monitoring point (Figure 3C). This data indicated that IGF2BP1 strongly enhances DHAV-1 IRES-mediated translation efficiency.

Next, it was wondered whether IGF2BP1 enhances DHAV-1 translation efficiency through binding to 3′ UTR of DHAV-1. The same copies of pDHAV-3′UTR-ΔA_25_, pDHAV-Δ3′UTR-A_25_, and pDHAV-Δ3′UTR-ΔA_25_ were transfected into the ExIGF2BP1 and SiIGF2BP1 DEE cells, and the FLuc activity was measured at 4 hpt. In the pDHAV-3′UTR-ΔA_25_ group, FLuc activity significantly (*p* < 0.001) increased in the ExIGF2BP1 DEE cells or significantly (*p* < 0.001) decreased in the SiIGF2BP1 DEE cells (Figure 3D). In the pDHAV-Δ3′UTR-A_25_ group (absent of 3′ UTR), the expression level of IGF2BP1 does not affect the translation efficiency of DHAV-1 (Figure 3D). This data indicated that the interaction between IGF2BP1 and 3′ UTR was essential for enhancing DHAV-1 translation efficiency. 

### 3.3. Viral Protein Accumulation Significantly Increases IGF2BP1 Expression Level

To identify the correlation between viral propagation and IGF2BP1 expression, the RNA-launched infectious clone pR-DHAV-1 was transfected into DEE cells. Cell lysates were collected from 12 hpt to 48 hpt to measure viral genome and protein increment level. The result showed that DHAV-1 propagated efficiently in DEE cells and that viral propagation significantly increased IGF2BP1 expression levels in the DEE cells transfected with pR-DHAV-1 (Figure 4A). IFA result also intuitively showed that viral propagation enhances IGF2BP1 expression level (Figure 4B).

Next, the plasmids pBluescript-polyprotein (Figure 1E) and pBluescript II KS (+) empty vector were transfected into DEE cells. Western blot assay showed that viral protein accumulation significantly enhanced IGF2BP1 expression level while transfection of the pBluescript II KS (+) vector had no impact on IGF2BP1 expression (Figure 4C).

In order to find out whether the DHAV-1 viral genome accumulation without viral proteins affects IGF2BP1 expression level, the in vitro transcribed product of pBlue-Ribo-DHAV-1 (Figure 1D) was transfected into DEE cells. The result showed that DHAV-1 viral genome reached up to 10^3.73^ copies/µl at 1 hpt, while viral protein was undetectable (Figure 4D). Clearly, IGF2BP1 expression level exhibited no obvious difference (*p* > 0.05) in the DEE cells with different levels of viral genome accumulation but no viral protein accumulation (Figure 4D).

### 3.4. IGF2BP1 Does Not Affect DHAV-1 Replication

To determine whether IGF2BP1 affected DHAV-1 replication, the same amount of the in vitro transcribed RNAs of the pR-DHAV-1 were transfected into ExIGF2BP1 DEE, SiIGF2BP1 DEE, and normal DEE cells. The viral copy numbers in the supernatant and cells were determined from 12 hpt to 60 hpt by RT-qPCR. The results showed that no significant difference (*p* > 0.05) on viral copy number was observed at each time point in the normal DEE, SiIGF2BP1 DEE, and ExIGF2BP1 DEE cells (Figure 5). These data indicated that IGF2BP1 had no impact on DHAV-1 replication.

## 4. Discussion

Most RNA viruses have evolved strategies to regulate cellular translation in order to promote preferential expression of the viral genome [30]. Efficient mRNA translation requires a series of protein–mRNA and protein–protein interactions [31]. Structural elements of the DHAV-1 viral mRNA, including the IRES element within the 5′ UTR, 3′ UTR, and poly(A) tail, are important determinants of these interactions. Currently, little of these viral structures are known to interact with host proteins, particularly with the host translation machinery, and to thus regulate the viral propagation. In this research, employing RNA–protein interaction techniques, we showed that IGF2BP1 specifically interacted with the DHAV-1 3′ UTR (Figure 2). To investigate the specificity of the IGF2BP1 function in DHAV-1 IRES-mediated translation, the lentiviral system and RNAi against IGF2BP1 was performed in DEE cells. With the increment expression level of IGF2BP1 in DEE cells, the IRES-mediated translation showed a significant increase (Figure 3C). While the control group had no effect on DHAV-1 IRES-mediated translation, a reduction of IGF2BP1 did result in a specific decrease of DHAV-1 IRES-mediated translation efficiency (Figure 3C). This effect was magnified in living cells when the 3′ end sequences were placed in a chimeric bicistronic construct, in which the DHAV-1 IRES promoted internal translation initiation of the second cistron (Figure 3B). With the increment of IGF2BP1 expression level to 289%, the IRES-mediated translation efficiency increased 34.3% or 29–49% in the reporter RNA group or the RNA-launched infectious clone group, respectively (Figure 3B,C). When the IGF2BP1 expression level was downregulated to 31%, the IRES-mediated translation efficiency showed a significant decrease of 66.7% or 59–67% in the reporter RNA group or the RNA-launched infectious clone group, respectively (Figure 3B,C). Next, we investigated whether IGF2BP1 enhances DHAV-1 translation efficiency via binding to 3′ UTR. The Fluc activity of reporter RNA that possessed 3′ UTR (pDHAV-3′UTR-A_25_ and pDHAV-3′UTR-ΔA_25_) was significantly affected by the IGF2BP1 expression level (Figure 3B,D), while Fluc activity of reporter RNA absent of 3′ UTR (pDHAV-Δ3′UTR-A_25_ and pDHAV-Δ3′UTR-ΔA_25_) remain unaffected in DEE cells with different IGF2BP1 expression levels (Figure 3D). These collected data indicated that IGF2BP1 strongly enhanced IRES-mediated translation efficiency via binding to DHAV-1 3′ UTR.

Up until now, the relationship between IGF2BP1 expression level and picornaviruses propagation is still unclear. Here, our study showed that the viral proteins in cytoplasm were required for the increment of IGF2BP1 (Figure 4A–C), while the viral genome accumulation alone had no impact on IGF2BP1 expression level (Figure 4D). The increased IGF2BP1 level might be attributed to the increased transcription of IGF2BP1 RNA, to posttranscriptional modifications of IGF2BP1 mRNA, and to prolonged protein half-life [32]. More importantly, the increment range in the polyprotein-expression group was lower compared to the RNA-launched infectious clone groups (Figure 4A,C). This might be due to the fact that the endogenous IGF2BP1 in DEE cells strongly enhanced viral protein expression progress via specifically binding to 3′ UTR in RNA-launched infectious clone group yet failed to stimulate the expression level in the polyprotein group that lacks the viral 3′ UTR. Thus, the viral protein express level in the infectious clone group was higher than that of the polyprotein group, which might lead to a higher increment degree of IGF2BP1 in the RNA-launched infectious clone group. In addition to its stimulated role in translation efficiency, it was wondered whether IGF2BP1 could also participate in the viral RNA replication process. Subsequent data showed that IGF2BP1 had no effect on DHAV-1 replication (Figure 5). However, viral replication levels in the SiIGF2BP1 group exhibited slightly (*p* > 0.05) lower levels compared to normal and ExIGF2BP1 groups. Further investigations are need for the mechanism of IGF2BP1 elevation.

Previous studies had shown that IGF2BP1 is expressed during the embryonic stage and that it is essential for early development [22,33]. Insulin-like growth factor-2 mRNA-binding protein-1-deficient mice show dwarfism, impaired gut development, and downregulated IGF2 expression level at the embryonic stage [33]. Except in some reproductive organs, IGF2BP1 was not expressed or at a low level in most postnatal organs of human, mouse, zebrafish, chicken, and mallard, but it was expressed in all organs in Pekin duck after hatching [34]. The VP1 of DHAV-1 played important roles in producing neutralizing antibodies and binding to cell surface specific receptors [35]. In the DHAV-1 infected ducklings, the increased viral protein expression derived by IGF2BP1 would stimulate the immune system to produce higher levels of neutralizing antibodies to resist DHAV-1 infection. In addition, as the level of protein expression increases and the level of genomic replication remains unchanged, the remaining viral capsid proteins might bind to cell surface receptors and inhibit DHAV-1 infection. The mutual promotion effect between IGF2BP1 and viral translation might be an active self-defense mechanism for young ducklings against DHAV-1 invasion.

In this study, we demonstrated that IGF2BP1 specifically interacted with 3′ UTR and significantly enhanced DHAV-1 IRES-mediated translation efficiency without affecting the replication of DHAV-1 (Figure 2, Figure 3, and Figure 5). Interestingly, DHAV-1 viral protein accumulation also remarkably increased IGF2BP1 expression level in DEE cells, whereas the accumulation of the DHAV-1 genome had no impact on IGF2BP1 expression level (Figure 4). It would be interesting to investigate the role of the mutual promotion effect between IGF2BP1 and viral translation in young ducklings against DHAV-1 invasion.

## 5. Conclusions

In the present study, we first found that IGF2BP1 specifically interacted with the DHAV-1 3′ UTR by RNA pull-down assay. The interaction between IGF2BP1 and DHAV-1 3′ UTR strongly enhanced IRES-mediated translation efficiency but failed to regulate DHAV-1 replication in DEE cells. The viral propagation of DHAV-1 strongly enhanced IGF2BP1 expression level, and viral protein accumulation was the key point to this increment. Our data demonstrated the positive role of IGF2BP1 in DHAV-1 viral proteins translation and provided data support for the replication mechanism of the virus.

## Figures and Tables

**Figure 1 biomolecules-09-00594-f001:**
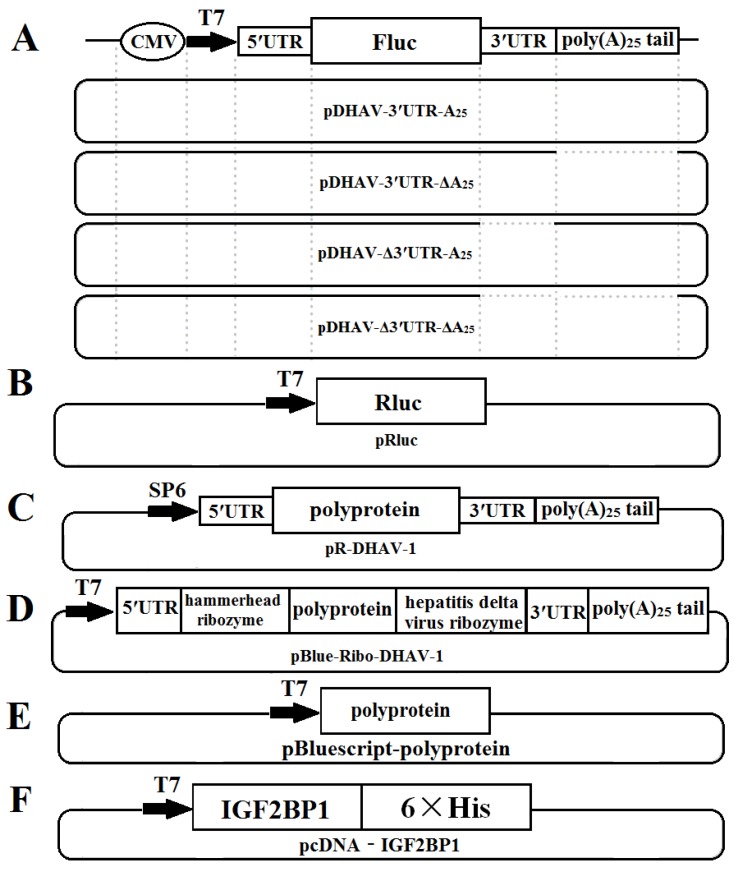
Construction schematic of the plasmids used in this study: (**A**) Construction of monocistronic reporter plasmids; (**B**) construction of plasmid pRluc; (**C**) construction of DHAV-1 RNA-launched infectious clones; (**D**) construction of plasmid pBlue-Ribo-DHAV-1; (**E**) construction of plasmid pBluescript-polyprotein; (**E**) construction of plasmid pBlue-Ribo-DHAV-1; and (**F**) construction of plasmid pcDNA-IGF2BP1.

**Figure 2 biomolecules-09-00594-f002:**
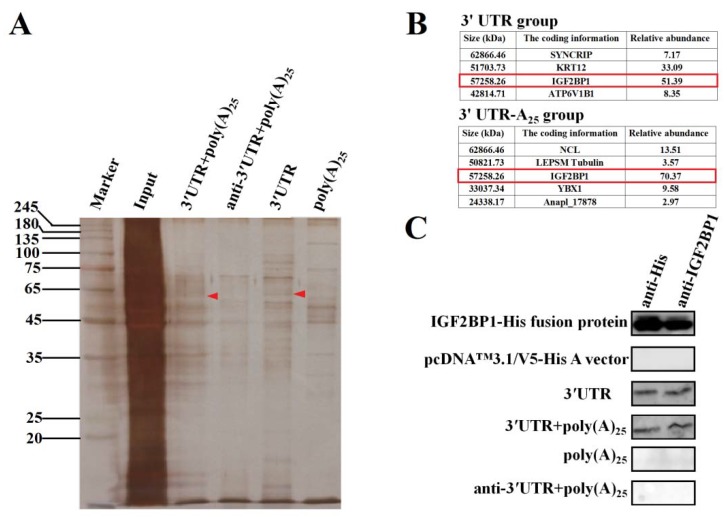
Insulin-like growth factor-2 mRNA-binding protein-1 (IGF2BP1) specifically binds to the 3′ untranslated region (UTR) of duck hepatitis A virus type 1 (DHAV-1). (**A**) Silver-staining electrophoresis of the proteins that specifically bind to the four labeled RNA probes: the entire 3′ UTR of DHAV-1 with poly(A)_25_ tail (probe 1), the reverse complementary sequence of the 3′ UTR of DHAV-1 with poly(A)_25_ tail (probe 2), the 3′ UTR of DHAV-1 alone (probe 3), and the poly(A)_25_ tail alone (probe 4). The red arrows showed the proteins interacted with probes 1 and 3 rather than the probes 2 and 4. (**B**) The Matrix-assisted laser desorption/ionization time-of-flight mass spectrometry (MALDI-TOF-MS) results of the proteins marked with red arrows in Figure 2A. IGF2BP1 was a common protein that specifically interacted with the probes 1 and 3 (marked with red frames). (**C**) The purified IGF2BP1-His fusion proteins and in vitro pulling-down proteins with the four RNA probes were conducted by western blot assay using anti-IGF2BP1 or anti-His antibody (1:3000).

**Figure 3 biomolecules-09-00594-f003:**
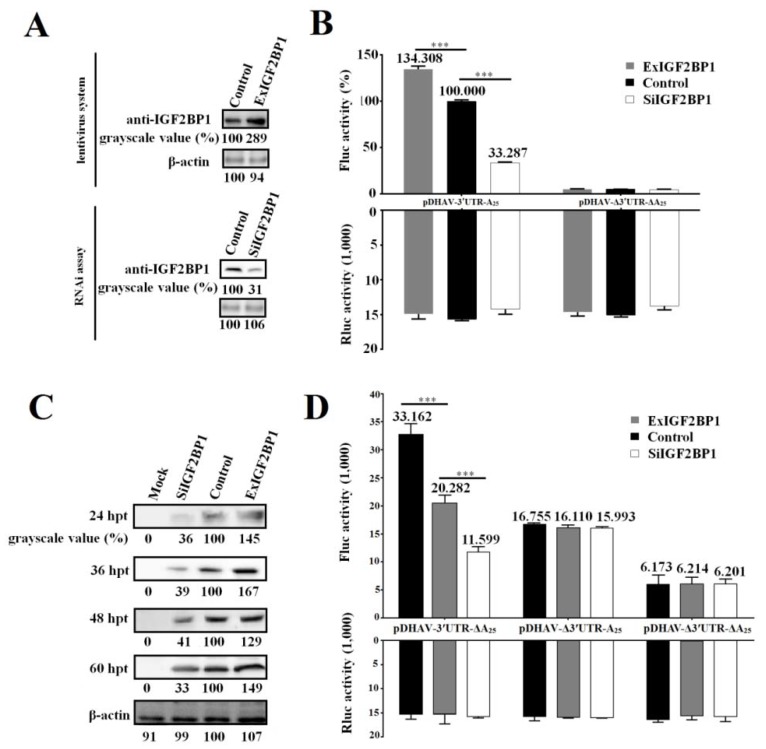
IGF2BP1 strongly enhances DHAV-1 internal ribosome entry site (IRES)-mediated translation efficiency via binding to 3′ UTR. (**A**) The IGF2BP1 expression level in ExIGF2BP1 duck embryo epithelial (DEE) and SiIGF2BP1 DEE cells was tested by western blot assay with anti-IGF2BP1 antibody. (**B**) The same quantity (0.8 µg) of the in vitro transcribed products of linearized plasmids pDHAV-3′UTR-A_25_ and pDHAV-Δ3′UTR-ΔA_25_ was transfected into the ExIGF2BP1 DEE, SiIGF2BP1 DEE, and normal DEE cells, and the luciferase activities were detected at 4 hpt. Reporter RNA of pRluc was transfected into the DEE cell line as the control. The FLuc activity of pDHAV-3′UTR-A_25_ in the normal DEE cell line was set as 100%. (**C**) The RNA-launched infectious clone pR-DHAV-1 was transfected into the ExIGF2BP1 DEE, SiIGF2BP1 DEE, and normal DEE cells, and the cell lysates were collected from 24 hpt to 60 hpt for western blot analysis with anti-DHAV-1 mAb 4F8. The grayscale value of the normal DEE cells was set as 100%. Columns and bars represent means of three independent transfections and standard deviations. (**D**) The same quantity (0.8 µg) of the in vitro transcribed products of linearized plasmids pDHAV-3′UTR-ΔA_25_, pDHAV-Δ3′UTR-A_25_, and pDHAV-Δ3′UTR-ΔA_25_ were transfected into the ExIGF2BP1 DEE, siIGF2BP1 DEE, and normal DEE cells, respectively, and the FLuc activity was detected at 4 hpt. The reporter RNA pRluc were transfected into DEE cells as the control. ****p* < 0.001.

**Figure 4 biomolecules-09-00594-f004:**
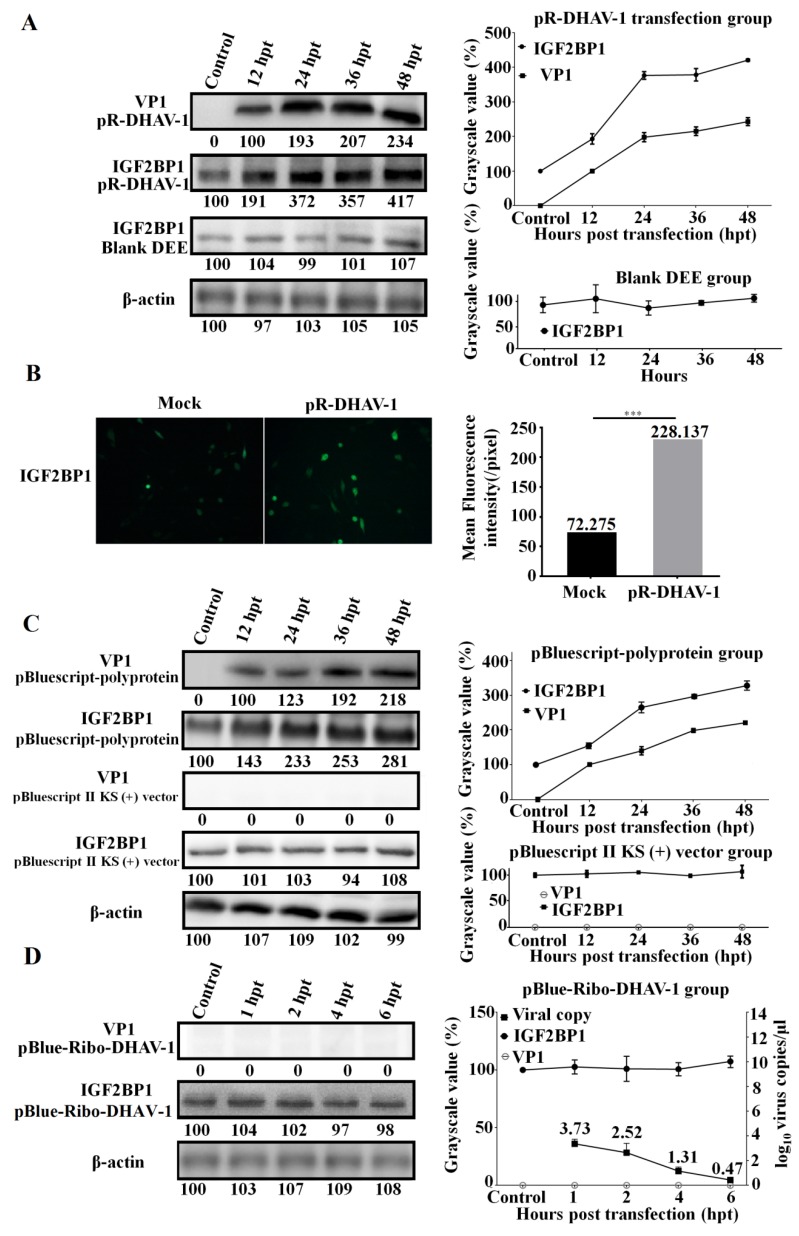
Viral propagation significantly enhanced IGF2BP1 expression level. (**A**) The DEE cells were transfected with 0.8 µg of the in vitro transcribed RNA of RNA-launched infectious clone pR-DHAV-1 and were collected from 12 hpt to 48 hpt for western blot assay with anti-DHAV-1 mAb 4F8 and anti-IGF2BP1 antibody, respectively (Left). The variation tendency of the VP1 of DHAV-1 and IGF2BP1 in the transfected DEE cells was measured from 12 to 48 hpt (Right). (**B**) The IGF2BP1 expression level in normal DEE cells (Mock group) and DEE cells transfected with 0.8 µg of pR-DHAV-1 (pR-DHAV-1 group) was measured by immunofluorescence assays (IFA) at 24 hpt with anti-IGF2BP1 antibody. The mean fluorescence intensity was measured by Image J software (National Institutes of Health, Maryland, USA). (**C**) The DEE cells were transfected with equal copies of the recombinant plasmid pBluescript-polyprotein and pBluescript II KS (+) vector, and the VP1 of DHAV-1 and IGF2BP1 expression levels were detected by western blot from 12 to 48 hpt (Left). The relationship between VP1 of DHAV-1 increment level and IGF2BP1 expression level in the transfected DEE cells was compared (Right). (**D**) The DEE cells were transfected with the in vitro transcribed RNA of pBlue-Ribo-DHAV-1, and the VP1 of DHAV-1 and IGF2BP1 expression levels were detected by western blot at 1, 2, 4, and 6 hpt (Left). Total RNA was extracted from collected DEE cells and was immediately used for detection of DHAV-1 by RT-qPCR. The relationship between the viral genome copy numbers of DHAV-1 and IGF2BP1 expression level in the transfected DEE cells was compared (Right). Bars represent the mean ± standard deviation of three replicate experiments for each group.

**Figure 5 biomolecules-09-00594-f005:**
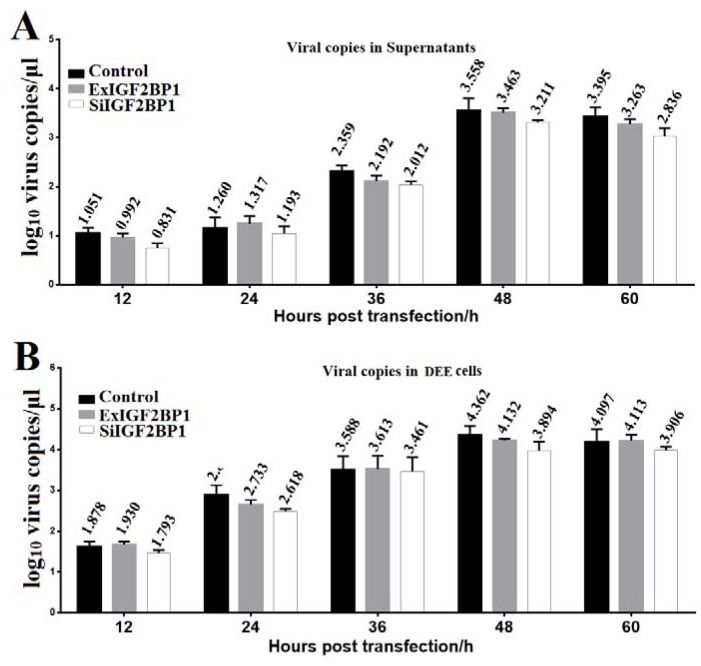
IGF2BP1 had no impact on viral replication. Equal amounts of the in vitro transcribed RNA of RNA-launched infectious clone pR-DHAV-1 were transfected into ExIGF2BP1 DEE, SiIGF2BP1 DEE, and normal DEE cells. Total RNA was extracted from the supernatant (**A**) and DEE cells (**B**), and the extracted RNA was immediately used for RT-qPCR measurement. Bars represent the mean ± standard deviation of three replicate experiments for each group.

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
