# Peer review of "IGF2BP1 Significantly Enhances Translation Efficiency of Duck Hepatitis A Virus Type 1 without Affecting Viral Replication"

_biomolecules, 2019, doi:10.3390/biom9100594_

Round 1
Reviewer 1 Report
Minor correct of English
Reviewer 2 Report
The article "IGF2BP1 significantly enhance translation efficiency of duck hepatitis A virus type 1 without affecting viral replication" by Chen et al. identified IGF2BP1 is one of the potential proteins that interact with DHAV‐1 3′ UTR. They also analyzed the role of IGF2BP1 in DHAV‐1 translation and replication. Overall, this is interesting report. But I would like to raise some major concerns.
Pull-down assay using total cell lysate is not a sufficient experiment to support the interaction specificity. The observed IGF2BP1-RNA interaction may be mediated via an unknown RNA-binding protein in the cell lysate. In order to prove the direct interaction, the authors should use the purified (recombinant) protein and the RNA probe, and do experiments such as an electrophoretic mobility shift assay (EMSA) or in vitro pull-down assay. In the abstract, the authors stated “Collectively, our data demonstrated the positive role of IGF2BP1 in DHAV-1 viral proteins translation and provided new insights for the replication mechanism of DHAV-1”. However, I think the data regarding to the replication mechanism is not enough for providing insights. The authors should explain the role of 3′ UTR in DHAV replication (such as your previous finding published in Front. Microbiol. 2018, 9, 2250) in the introduction section. Also, they should discuss more in detail in the discussion section. Otherwise, they should rephrase this sentence. Please discuss the lack of the genome replication result in SiIGF2BP1 DEE in the figure 5. Can you compare the sequence of DHAV‐1 3′ UTR with the 3′ UTR sequences from other related virus? If IGF2BP1 really binds DHAV‐1 3′ UTR, can you identified the zipcode RNA element in DHAV‐1 3′ UTR by analyzing the multiple sequence alignment of 3′ UTR sequences? This would strengthen your conclusions significantly. (see for example Cell Reports 18, 1187–1199, January 31, 2017)
Minor revision
In the materials and methods section, the centrifugal force g should be italicized. Restriction enzyme names should not include a space between the main acronym and the Roman numeral. “Xho I” should be “XhoI”. The font size in figures 1-4 is too small for reading. The resolution of figure 2B should be improved.
Reviewer 3 Report
The authors have identified and demonstrated that insulin‐like growth factor-2 mRNA‐binding protein‐1(IGF2BP1) specifically interacted with 3′ untranslated region (3′ UTR) of DHAV-1 and enhanced IRES-mediated translation. This is an interesting story but was not well-written. Language editing is needed.
It seems to me there are many proteins were pulled down in Fig. 2a, it is not clear why the authors focused only IGF2BP1 bands. In this manuscript the authors mainly reported phenomena but lack of mechanism studies. For example, the authors claimed that “Viral protein accumulation significantly increases IGF2BP1 expression level”, but did not explore why the accumulation of DHAV polyprotein increases IGF2BP1 expression? Whether infection of DHAV-1 will increase the expression of IGF2BP1? Is this phenomena cell dependent (only DEE cell line)? It seems to me there is logical problems with sections 3.2 and 3.3. Since IGF2BP1 enhances IRES-mediated translation of DHAV-1 and the accumulation of DHAV-1 polyprotein increases IGF2BP1 expression, I am wondering where is the end during DHAV-1 infection? The cells will be full of polyprotein proteins and IGF2BP1? The conclusion in section 3.3 is not well-supported by the data. Based on section 2.10, plasmid pR-DHAV-1 were used for DEE cell transfection, which is not consistent with section 3.3 saying “in vitro transcribed RNAs of the pR-DHAV-1” were used. Anyway, the authors concluded that “IGF2BP1 does not affect DHAV‐1 genome replication” by measuring the virus titer (copy # of viral genome). I would say IGF2BP1 does not affect DHAV‐1 replication rather than genome replication. It is possible that IGF2BP1 might decrease genome replication but increase virus assembly, therefore, giving the same virus titer.
Minor comments:
Section 2.5, please check the lentiviral system used in current study. 3A and 3B can be combined. The same thing for Fig. 3C and 3E. Line 239 “In order to study the specific role of IGF2BP1 in DHAV-1 replication process,… ” I don’t think the data presented in this paragraph are directly related with “DHAV-1 replication”. Line 269, there is no information about “pBluescript‐polyprotein”, which should be included. Fig. 4B, the data from single cell might be not strong enough to support your statements. More cells should be included in single field.
Round 2
Reviewer 2 Report
Major comments:
1. pcDNA‐IGF2BP1 information is not enough. Please indicate the location of His-tag in the recombinant IGF2BP1 (N-terminal or C-terminal fusion?). You can make a diagram in Fig.1 or state that in section 2.2.
2. The vendor information of anti-His antibody is missing in section 2.9.
3. It would be better if the authors provide an SDS-PAGE image of the purified recombinant IGF2BP1 (stained by silver or coomassie blue) in Fig.2. to role out the possibility of impurity.
Minor comments:
1. Fig.2A: please enlarge this PAGE in 2X.
2. Fig.2A: replace the red arrows with red triangles with a bigger size.
3. Page 3 line 110: poly(A)25 where 25 should be subscripted.
Reviewer 3 Report
I have no further comments
